# Preparation of Wax-Based Warm Mixture Additives from Waste Polypropylene (PP) Plastic and Their Effects on the Properties of Modified Asphalt

**DOI:** 10.3390/ma15124346

**Published:** 2022-06-20

**Authors:** Gang Zhou, Chuanqiang Li, Haobo Wang, Wei Zeng, Tianqing Ling, Lin Jiang, Rukai Li, Qizheng Liu, Ying Cheng, Dan Zhou

**Affiliations:** 1School of Civil Engineering, Chongqing Jiaotong University, Chongqing 400074, China; cjhs_2000@163.com (G.Z.); lingtq@163.com (T.L.); jianglin_cn@foxmail.com (L.J.); lirukai89@163.com (R.L.); liuqizheng_cn@163.com (Q.L.); chengying_cn@163.com (Y.C.); zhoudan_cn1@163.com (D.Z.); 2School of Materials Science and Engineering, Chongqing Jiaotong University, Chongqing 400074, China; wanghaobopostbox@163.com (H.W.); lzzengwei@163.com (W.Z.)

**Keywords:** warm mix additive, waste polypropylene plastic, pyrolysis, PP wax, modified asphalt

## Abstract

The production of high-performance, low-cost warm mix additives (WMa) for matrix asphalt remains a challenge. The pyrolysis method was employed to prepare wax-based WMa using waste polypropylene plastic (WPP) as the raw material in this study. Penetration, softening point, ductility, rotational viscosity, and dynamic shear rheological tests were performed to determine the physical and rheological properties of the modified asphalt. The adhesion properties were characterized using the surface free energy (SFE) method. We proved that the pyrolysis temperature and pressure play a synergistic role in the production of wax-based WMa from WPPs. The product prepared at 380 °C and 1.0 MPa (**380-1.0**) can improve the penetration of matrix asphalt by 61% and reduce the viscosity (135 °C) of matrix asphalt by 48.6%. Furthermore, the modified asphalt shows favorable elasticity, rutting resistance, and adhesion properties; thus, it serves as a promising WMa for asphalt binders.

## 1. Introduction

In the asphalt pavement industry, the warm mix asphalt (WMA) has been introduced and used instead of the hot mix asphalt (HMA) to reduce the temperatures of mixture production and placement. This has led to reductions in fuel consumption and noxious gas emissions, which are particularly welcomed in urban environments [1,2,3,4]. A variety of asphalt binders and mixture additives are available to produce WMA mixtures among which wax-based additives are common types. For example, Sasobit can effectively reduce the viscosity of asphalt binder at high temperatures, thereby decreasing the mixing and compaction temperatures and improving the asphalt rutting resistance at usage temperatures [5,6,7,8,9]. Polythene wax (PEW) and polypropylene wax (PPW) are two other wax-based additives. They can also effectively reduce the construction temperature of asphalt mixtures [10,11,12]. However, the types of warm mixtures are still very limited, and the cost of existing warm mixes is high [13]; therefore, it is necessary to develop new low-cost warm mix additives (WMa). The production of WMa from waste materials (e.g., waste plastic) is an attractive approach.

Plastic products are essential in various fields owing to their low cost, light weight, and durability. The plastics are generally designed to be robust and durable, leading to their inherent non-degradable nature. Plastic wastes in the environment are of increasing concern because of their effects on wildlife and human. Thus, the treatment of plastic wastes has become a huge problem [14]. Most plastic wastes fall into four categories: polyester, polyolefin, polyvinyl chloride (PVC), and polystyrene (PS). Polyolefin, such as polyethylene (PE) and polypropylene (PP), with production of about 218 Mt every year, accounts for 57% of the plastic content of municipal solid waste [15]. As one of the representatives of plastics, PP plastic is extensively used due to its abundant source, low price, and chemical corrosion [16]. The widespread use of PP plastics has caused serious pollution. Therefore, there is an urgent need for the effective and environmentally friendly management of waste PP plastics [17]. Typical methods used to discard waste plastic include landfills, incineration, and mechanical recycling. However, landfill and incineration methods have the potential to pollute the surrounding soil, air and water [18]. Furthermore, the mechanical recycling method degrades the physical properties of recycled PP plastic [19]. Because of its high-hydrocarbon content, the use of waste PP plastic for the production of high-value-added products and energy recovery has attracted extensive attention among researchers, as it can be easily transformed to alkenes and alkanes via pyrolysis [17]. Because the wax can be used as WMa or can be used for improving the thermal stability of asphalt binder [20,21], the production of PP wax (PPW) should be an excellent strategy for the resource utilization of waste PP plastic. Current research has focused on the production of fuel from waste plastics via thermal pyrolysis and catalytic cracking [22,23,24]. Few studies have been devoted to the production of waxes from waste PP plastic, particularly those used to modify asphalt. Therefore, it is necessary to conduct a systematic study on the pyrolysis of waste PP plastics for wax production and study in depth the effect of wax on the performance of modified asphalt. In addition, few studies have been conducted on the combined effects of temperature and pressure on the pyrolysis of waste PP plastics.

In this study, a range of PPW samples was prepared by pyrolysis at different temperatures and pressures. Furthermore, the physical, viscosity-temperature, and rheological properties of the PPW-modified asphalt binders were analyzed. In addition, the surface free energy (SFE) method was used to characterize the adhesive properties of the PPW-modified binders. The results showed that the PPWs obtained at 380 ≤ T ≤ 400 °C and 0.5 ≤ *p* ≤ 1.0 MPa exhibited good warm mixing performances. To the best of our knowledge, this is the first study to investigate the combined effects of temperature and pressure on preparing WMa from waste PP plastic.

## 2. Materials and Methods

### 2.1. Materials

Waste PP plastic particles (WPPs) were obtained from a plastic recycling plant. Matrix asphalt (MA) with a penetration grade of 70 (penetration at 25 °C is 6.65 mm; softening point is 48.2 °C; ductility at 15 °C is greater than 100 cm; glass transition temperature (T_g_) is around −10 °C) was used in this work. Ethylene glycol (99+%), formamide (99+%), and deionized water were used as liquid solvents for the measurement of SFE components. All materials were of commercial origin and were used without further purification.

### 2.2. PPW Preparation

PPW was synthesized via pyrolysis in a pressure reactor (Figure 1). The WPP was placed in the pressure reactor, and the reactor was sealed and purged with nitrogen for 5 min to discharge the air in it. The air release valve was closed, nitrogen was injected at a specified pressure, and the air intake valve was closed. The reactor was heated to a specified temperature and then stirred at 120 revolutions per minute (rpm) for 20 min. During this period, the pressure in the reactor was maintained constant by adjusting the pressure-relief valve. Subsequently, the reactor was cooled to 120 °C, and the target PPW was poured out and collected in zip-lock bags.

### 2.3. Preparation of PPW-Modified Asphalt (PPWA)

The PPWA was prepared by blending PPW and an asphalt binder using a high-speed shearing machine at 120 °C and 1000 rpm for 10 min. The dosage of PPW was 6% of the weight of the matrix asphalt binder. The synthetic conditions for PPW and the corresponding modified asphalt are listed in Table 1.

### 2.4. Binder Tests

Penetration [25], softening point [26], and ductility tests [27] were performed to evaluate the conventional physical properties of PPWA and MA. Penetration tests were performed at three temperatures (5 °C, 10 °C, and 15 °C) to calculate the penetration index (PI) of the binder.

To identify the rotational viscosities of the binders, 10 g of MA or PPWA samples were tested using an NJD-1D Brookfield viscometer (Shanghai Changji Geological Instrument Co. Ltd., Shanghai, China) [28]. Viscosity tests were conducted from 105 to 165 °C at 15 °C intervals.

The viscoelastic properties of the binders were characterized using a dynamic shear rheometer (DSR) according to the TA DHR-3 (TA Instruments Inc., New Castle, DE, USA). The high- and intermediate-temperature performances were respectively characterized using superpave rutting and fatigue parameters [29]. Plates with specific characteristics (gap: 1 mm, diameter: 25 mm) were used for each binder to obtain the DSR value. The complex modulus (G*) and phase angle (δ) were recorded for rheological analysis.

The adhesion property of asphalt binder with aggregate was evaluated based on the SFE theory [30,31,32]. To calculate the surface energy, the contact angles between the asphalt and the three liquid solvents (distilled water, glycerol, and formamide) were measured using the Wilhelmy plate method [33]. First, the asphalt sample was heated to the melted state, and the cleaned glass slide was then immersed in hot asphalt for a few minutes; then, it was removed so that a layer of asphalt film with a uniform thickness was formed on the surface of the glass slide (Figure 2a). The contact angles of the three liquids with the asphalt film were measured at 20 °C using a HARKE-SPCAX3 contact-angle meter (Beijing Harke Test Instrument Factory, Beijing, China, Figure 2b). The experiment was repeated five times, and the average values were obtained.

The microstructure of PPW was tested by a DM2500 optical microscope (Leica, Wetzlar, Germany). FTIR spectra of PPW and asphalt binders were tested on the Tensor Ⅱ spectrometer (Bruker, Germany). DSC measurements were performed on a DSC214 System-Instrument (Netzsch Company, Selb, Germany).

## 3. Results and Discussion

### 3.1. Softening Point

The softening points of MA and PPWA are shown in Figure 3. It can be observed that the incorporation of PPW increased the softening point of MA, thus suggesting an improvement of the temperature stability of asphalt [1]. By comparing the PPWAs of **A** to **H**, it can be observed that at the same pyrolysis pressure, with an increase in pyrolysis temperature, the softening point of the obtained PPWA decreased; at the same pyrolysis temperature, the softening point of the obtained PPWA also decreased as a function of pyrolysis pressure. This indicates that an increased number of small molecules may be produced with an increase in pyrolysis temperature or pressure. However, the softening point of sample **I** is slightly higher than that of **E**, **F** and **H**. Moreover, the softening points of the four samples are not much different. This behavior suggests that increasing the pressure during pyrolysis at high temperature may have less effect on the high-temperature property of the products.

### 3.2. Penetration

Penetration can indicate degree of softness and the relative viscosity of asphalt binder. The higher the penetration value, the softer the asphalt binder and the lower the viscosity of asphalt [33]. Figure 4a shows the penetration values of MA and PPWA at 25 °C. It can be observed from the figure that the penetrations of **E**, **F**, **H**, and **I** are greater than that of MA, thus indicating that the corresponding PPWs can reduce the viscosity of MA. According to a previous study, the incorporation of a warm mix additive can reduce the viscosity of asphalt [34]. From this point of view, these four PPW, **380-0.5**, **380-1.0**, **400-0.5**, **400-1.0** exhibit the potential to be used as WMa. Furthermore, **380-1.0** seems to be the best one, and it can lead to 61% penetration improvement for MA. Furthermore, the penetration of **A**, **B**, **C**, **D**, **E**, and **F** increased sequentially, whereas those of **G**, **H**, and **I** first increased and then decreased. The results showed that the preparation conditions of PPW have a significant effect on the properties of PPWA. PPW produced by excessive temperature and pressure cannot reduce the viscosity of asphalt. This result is consistent with the softening point. This is an interesting phenomenon, and the reason remains to be investigated in the following chemical characterization.

The penetration index (PI) was calculated based on the penetration values of PPWA at different temperatures to evaluate the temperature sensitivity of the asphalt according to Equations (1) and (2). The results are shown in Figure 4b.
(1)lgP=K+AT
(2)$$PI=\frac{20-500A}{1+50A}$$
where *T* and *P* are the temperature and penetration, respectively, and *K* and *A* are determined from the lg*P*–*T* curve.

It is known that most asphalt binders have PI values in the range of −2 to +2. Additionally, as the PI value of asphalt increases, the temperature susceptibility decreases [35,36]. In addition, the flexibility and thixotropy of the asphalt binders increased as PI increased [33]. In this study, all PPWA have PI values in the range of −2 to +2, indicating that all PPWAs are sol–gel structures, which means that adding these PPWs does not affect negatively the temperature susceptibility. 

### 3.3. Ductility

Ductility can be used to evaluate the tensile deformation and flexibility of the asphalt binder. Figure 5 shows the ductility values of MA and PPWA at 15 °C. It can be observed that the ductility of MA decreases with the addition of PPW, thus indicating a negative effect of PPW on low-temperature flexibility and crack resistance of MA. Moreover, the ductility of PPWA increased as the corresponding pyrolysis pressure increased except for **I**. Among the nine PPWAs, **F** samples (the samples modified by PPW **380-0.1**) exhibited the best low-temperature flexibility, suggesting that the pyrolysis temperature and pressure are equally important in the production of PPW. Gao [37] proved that increasing the pressure at high pyrolysis temperature inhibited the degradation of macromolecular substances in polymers. Thereby, there might be more macromolecules in the PPW **400-1.0** than **400-0.5,** which may be the reason for the ductility of **I** sample lower than that of **H**.

### 3.4. Rotational Viscosity

Viscosity characterizes the ability of asphalt to resist shear deformation when subjected to an external force. Figure 6 shows the viscosity–temperature curves for MA and PPWA. It can be observed that the viscosities of **E**, **F**, **H**, and **I** are lower than that of MA. The addition of **380-0.5**, **380-1.0**, **400-0.5**, and **400-1.0** to MA led to a decrease in the viscosity, which reveals that the four PPWs can decrease the mixing and compaction temperatures of the asphalt mixture; accordingly, their use may improve workability at lower temperatures [33]. The most pronounced effect was observed for **F** (the sample modified by PPW **380-1.0**), which led to a viscosity reduction of 48.6% at 135 °C compared with MA. This result is comparable to the viscosity reduction for rubberized asphalt using commercial pure PPW [12]. It is expected that the wax-based WMa may have melted at lower temperatures, which may have caused a reduction in viscosity [38].

### 3.5. Rheological Properties

The rheological properties of MA and PPWA were investigated using a DSR. The complex modulus (G*) and phase angle (δ) master curves are shown in Figure 7 over the temperature range of 52–82 °C. As shown in Figure 7a, the G* values of **A**, **B**, **C**, and **D** are higher than those of MA, which implies that the addition of PPW **360-0.1**, **360-0.5**, **360-1.0** and **380-0.1** can improve the stiffness and flow deformation resistance of the asphalt binder [39]. Figure 7b shows that the addition of PPW **380-0.5**, **380-1.0**, **400-0.1**, **400-0.5**, and **400-1.0** promoted the decrease in G* of MA. This effect is probably due to the melting of PPW, thus leading to a change in the rheological behavior of PPWA [40,41]. Among the corresponding five PPWAs, **F** exhibited the highest G* value at all test temperatures, thus suggesting its relatively high resistance to shear deformation. This might be due to the PPW 380-1.0 exhibiting better crystallinity than the other four PPWs [40].

Figure 7c shows the curves of δ for MA and PPWA. The data show that PPW **360-0.1**, **360-0.5**, **360-1.0**, and **380-0.1** can decrease considerably the δ value of MA, thus indicating an improvement in the elasticity of asphalt [42]. Interestingly, the δ values of the corresponding four PPWAs first increased and then decreased as the temperature increased. This phenomenon may be attributed to the incomplete pyrolysis of PP at low temperatures and low pressures. Additionally, the PP particles remaining in the PPW improve the high-temperature elasticity of the binder [43]. As shown in Figure 7d, the four PPWs, **380-0.5**, **400-0.1**, **400-0.5**, and **400-1.0**, decreased the values of δ at low or high temperatures. A PPW of **380-1.0** can improve slightly the elasticity of MA. Furthermore, the influence of **380-1.0** on δ values yields a linear relationship with temperature, which is consistent with Sasobit [44].

The rutting factor (RF, G*/sinδ) is related to the ability of asphalt binders to resist rutting [39]. As shown in Figure 7c, the RFs of **A**, **B**, **C**, and **D** are much larger than that of MA, thus suggesting that the addition of the corresponding four PPWs improved the rutting resistance of MA. The PPW values of **380-0.5**, **380-1.0**, **400-0.1**, **400-0.5**, **400-1.0** slightly decreased the RF of MA. However, according to the superpave specification (AASHTO: MP1), the RF value should be at least 1.0 kPa for an asphalt binder at the maximum pavement design temperature [39]. The results in Figure 7f indicate that the RF values of **E**, **F**, **G**, **H**, and **I** are all greater than 1.0 kPa at a temperature of 64 °C. The value of the **380-1.0** modified asphalt **F** is the highest among the five PPWA, indicating the best resistance against rutting ability.

### 3.6. Adhesion Properties

By comparing the nine PPWA, it can be seen that the PPW 380-1.0 has the greatest potential as a WMa. However, the use of wax-based WMa is still limited due to the concern regarding the adhesion property of modified asphalt [11,45,46]. In this study, the adhesion property of MA and **F** was investigated based on SFE theory to ensure the workability of F asphalt compared to matrix asphalt. The limestone was used as the test aggregate. Adhesion energy (Δ*G_AS_*) and de-bonding energy (Δ*G_ASW_*) were calculated from Equations (3) and (4), respectively.
(3)$$\Delta G_{AS}=2\left(\sqrt{\gamma_{A}^{LW}\gamma_{S}^{LW}}+\sqrt{\gamma_{A}^{+}\gamma_{S}^{-}}+\sqrt{\gamma_{A}^{-}\gamma_{S}^{+}}\right)$$
(4)$$\Delta G_{ASW}=-2\left[\begin{array}{l} \sqrt{\gamma_{A}^{LW}\gamma_{W}^{LW}}+\sqrt{\gamma_{S}^{LW}\gamma_{W}^{LW}}-\sqrt{\gamma_{A}^{LW}\gamma_{S}^{LW}}-\gamma_{W}^{LW}\\ +\sqrt{\gamma_{W}^{+}}\left(\sqrt{\gamma_{A}^{-}}+\sqrt{\gamma_{S}^{-}}-\sqrt{\gamma_{W}^{-}}\right)\\ +\sqrt{\gamma_{W}^{-}}\left(\sqrt{\gamma_{A}^{+}}+\sqrt{\gamma_{S}^{+}}-\sqrt{\gamma_{W}^{+}}\right)\\ -\sqrt{\gamma_{A}^{+}\gamma_{S}^{-}}-\sqrt{\gamma_{A}^{-}\gamma_{S}^{+}} \end{array}\right]$$
where $\gamma_{A}^{LW}$/$\gamma_{S}^{LW}$/$\gamma_{W}^{LW}$, $\gamma_{A}^{+}$/$\gamma_{S}^{+}$/$\gamma_{W}^{+}$, $\gamma_{A}^{-}$/$\gamma_{S}^{-}$/$\gamma_{W}^{-}$ are the SFE nonpolar component, polar acid component, and polar base component of asphalt/aggregate/water.

The Compatibility Ratio (*CR*) is used to evaluate the moisture susceptibility of asphalt. A higher value of *CR* implies a lower release of free energy in the presence of moisture [47]. *CR* was calculated as
(5)$$CR=\frac{\Delta G_{AS}}{\left|\Delta G_{ASW}\right|}$$

The SFE parameters of the probe liquids and aggregate are shown in Table 2 and Table 3. The SFE components of limestone have been measured by Luo et al. [31]. The contact angles measured with each probe liquids are given in Table 4. The SFE parameters of asphalt can be calculated according to the Young–Dupre equation, and the results are given in Table 5. The calculated results of Δ*G_AS_*, │Δ*G_ASW_*│ and *CR* are shown in Figure 8. The results indicate that PPW **380-1.0** can increase the adhesion property and moisture susceptibility of MA. Wasiuddin et al. evaluated the adhesion property of Sasobit-modified asphalt using the SFE method [48]. The result showed that Sasobit can increase the cohesive strength slightly. From this point of this view, the PPW of **380-1.0** is similar to Sasobit.

### 3.7. Crystallinity and Composition of 380-1.0

The crystallinity of PPW 380-1.0 was measured by optical microscopy, as shown in Figure 9a. It can be seen that there are lots of crystals dispersed in PPW, which may be the reason for increasing the high-temperature performance for matrix asphalt [49]. Figure 9b shows the FTIR spectrum of 380-1.0. The absorption peaks at 2953, 2918, 1462, and 1377 cm^−1^ correspond to the -CH_2_ asymmetric and symmetric stretching bands. The peaks at 1462 and 1377 cm^−1^ correspond to the CH_3_ scissoring and symmetrical bending vibration. The peak at 876 cm^−1^ corresponds to the bending vibration of -CH=CH-. The FTIR results indicate that PPW 380-1.0 is mainly composed of alkanes and alkenes.

### 3.8. FTIR and DSC of PPWA F

The FTIR spectra of PPWA F and neat asphalt are shown in Figure 10a. The position of the main absorption bands of F and neat asphalt are not significantly different, which indicates that no obvious chemical reaction occurred in the modified bitumen after the addition of PPW. There are two significant exothermic peaks in the DSC curve of PPWA F. The peak at 20 °C corresponds to the phase translation of small molecules in modified asphalt. The peak at 100 °C can correspond to the melting of PPW in modified asphalt, because the melting point of matrix asphalt is generally higher than 150 °C. In addition, the T_g_ of PPWA F was also obtained from the DSC curve. The Tg of F is higher than that of MA, indicating that the PPW reduced the low-temperature performance of asphalt [50]. The result is consistent with the ductility result.

## 4. Conclusions

In summary, nine PPW were prepared via the pyrolysis approach at different temperature and pressure in a pressure reactor. The properties of PPW modified asphalt were characterized through penetration, softening point, ductility, viscosity and rheological tests. Moreover, the adhesion property of PPW **380-1.0** modified asphalt was investigated using the SFE method based on contact angle measurements. The main conclusions are as follows:(1)The softening point and ductility results show that the PPW prepared in this study can increase the high-temperature stability while decreasing the low-temperature flexibility of the asphalt.(2)The penetration and rotational viscosity results indicate that PPW of **380-0.5**, **380-1.0**, **400-0.5**, and **400-1.0** can reduce the viscosity of asphalt and exhibit the potential to be used as WMa. The corresponding four modified PPWAs have a PI value in the range of −1 to +1, thus suggesting that the addition of the four PPWs does not affect the temperature susceptibility.(3)The DSR results show that the addition of **360-0.1**, **360-0.5**, **360-1.0**, and **380-0.1** improves the flow deformation resistance and rutting resistance of the asphalt binder, whereas the other five PPWs did the opposite.(4)The PPW **380-1.0** reduced the viscosity of asphalt by 48.6% at 135 °C, and the corresponding PPWA **F** exhibited favorable elasticity and rutting resistance characteristics. In addition, **380-0.1** improved the adhesion property and moisture susceptibility of matrix asphalt, which was measured based on the SFE theory. The PPW **380-1.0** can be considered a promising warm mix additive for asphalt binders.(5)This study demonstrated that the temperature and pressure for the pyrolysis of PP play a synergistic role in the production of PPW-based WMa. The conditions of 380 ≤ T ≤ 400 °C and 0.5 ≤ *p* ≤ 1.0 MPa should be suitable.

The results obtained in this study show that the PPW prepared from waste PP plastic by the pyrolysis method can be used as WMa. Further research on the performance of PPW modified asphalt mixture is required. Furthermore, according to this work and previous research, the crystalline wax may have a negative influence on the low-temperature performance of asphalt binder [51,52]. So, the research focus is on developing the PPW WMa that can improve the low-temperature performance of asphalt, which should be carried out in the future.

## Figures and Tables

**Figure 1 materials-15-04346-f001:**
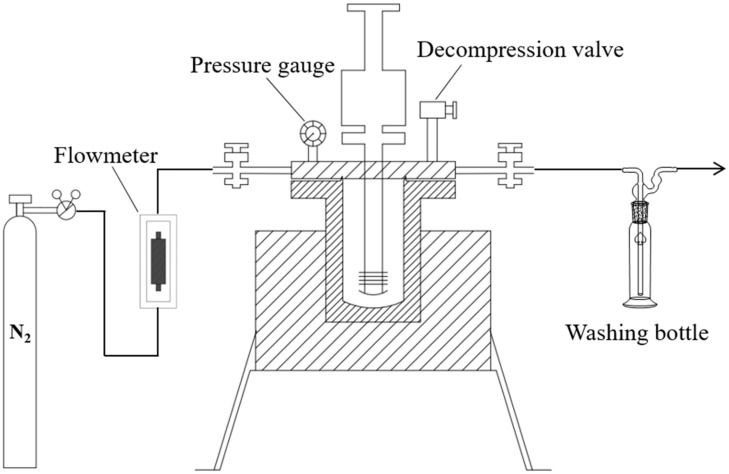
Schematic of pyrolysis reactor.

**Figure 2 materials-15-04346-f002:**
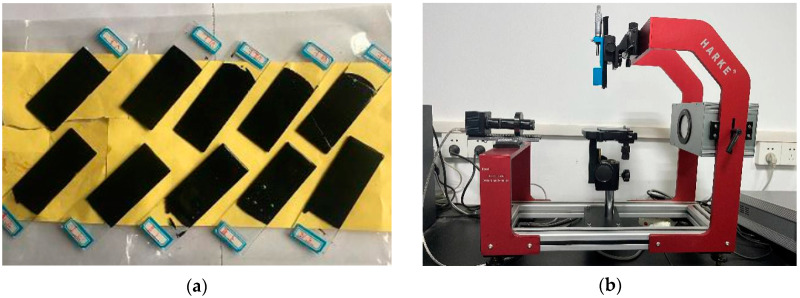
Asphalt film on glass slide (**a**) and contact-angle meter (**b**).

**Figure 3 materials-15-04346-f003:**
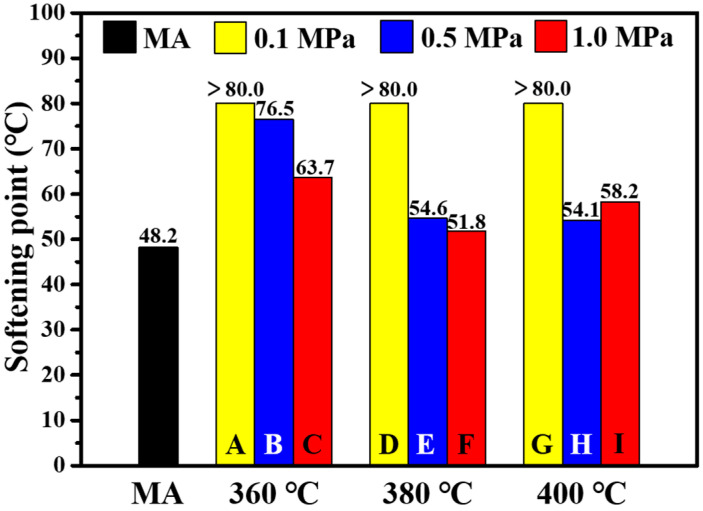
Softening points of matrix asphalt (MA) and PPW-modified asphalt (PPWA).

**Figure 4 materials-15-04346-f004:**
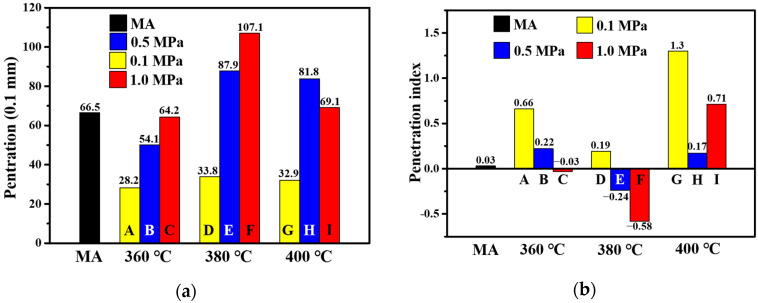
Penetration (**a**) and penetration index (**b**) of MA and PPWA.

**Figure 5 materials-15-04346-f005:**
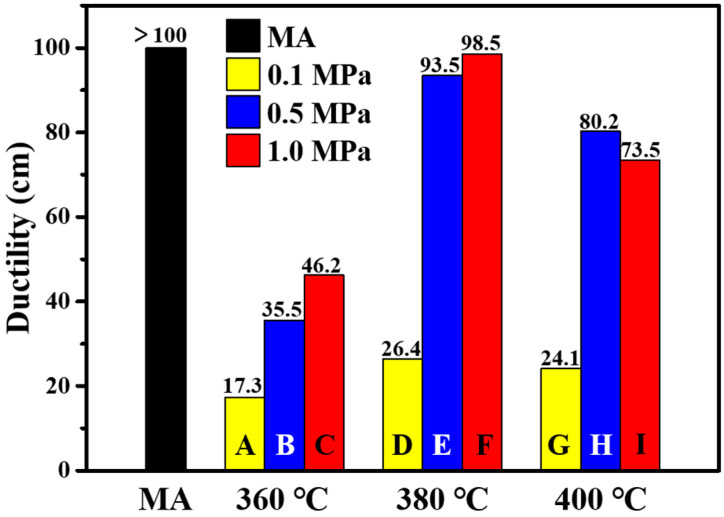
Ductility of MA and PPWA at 15 °C.

**Figure 6 materials-15-04346-f006:**
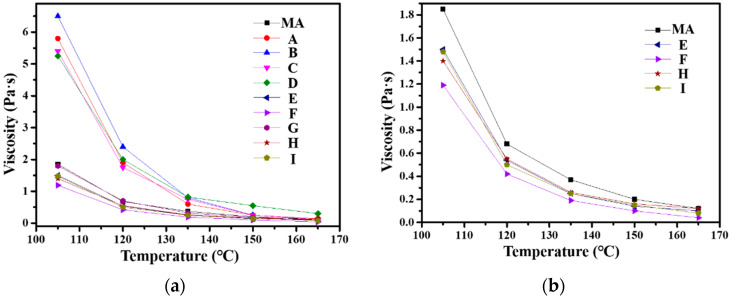
Viscosity–temperature curves of MA and PPWA (**a**). Magnified view of some of the curves (**b**).

**Figure 7 materials-15-04346-f007:**
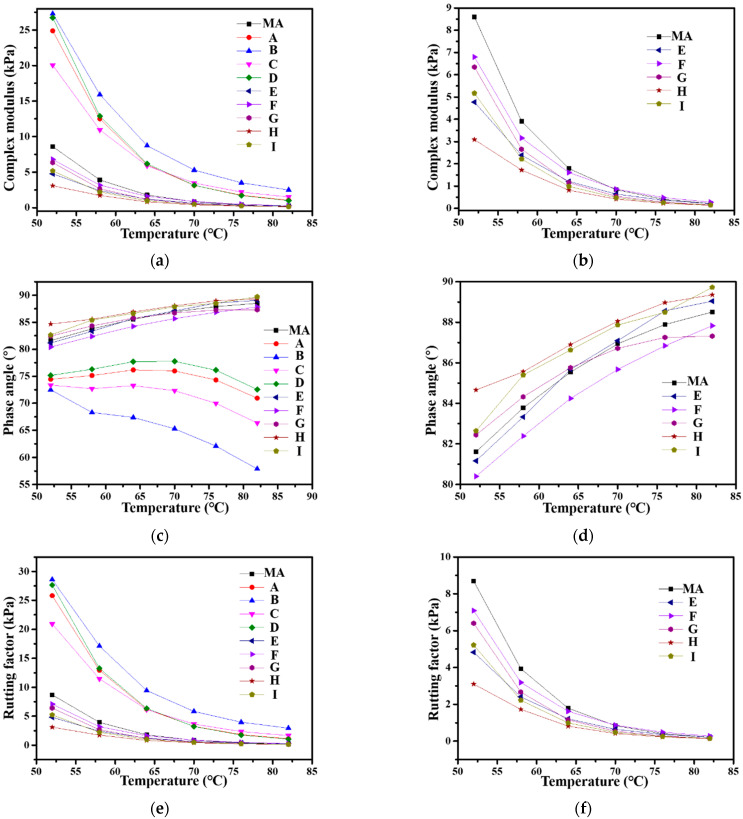
Rheological properties of MA and PPWA: (**a**,**b**) Complex modulus; (**c**,**d**) Phase angle; (**e**,**f**) Rutting factor.

**Figure 8 materials-15-04346-f008:**
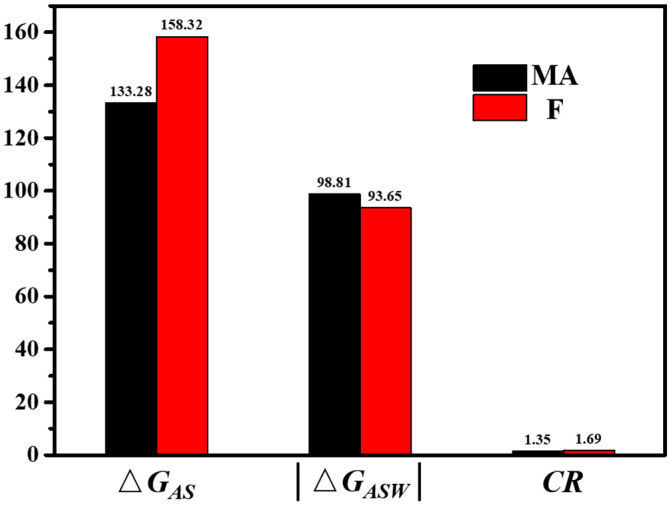
The Δ*G_AS_*, │Δ*G_ASW_*│ and *CR* of MA and F.

**Figure 9 materials-15-04346-f009:**
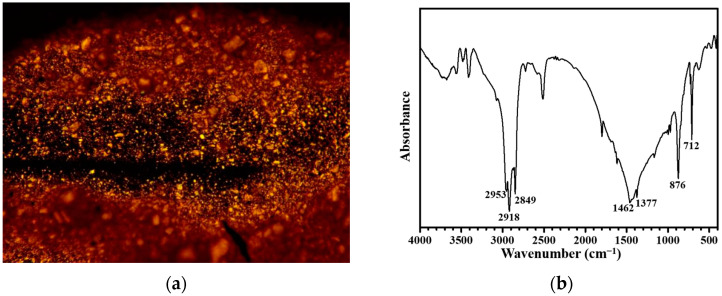
Microphotograph (**a**) and FTIR spectrum of PPW 380-1.0 (**b**).

**Figure 10 materials-15-04346-f010:**
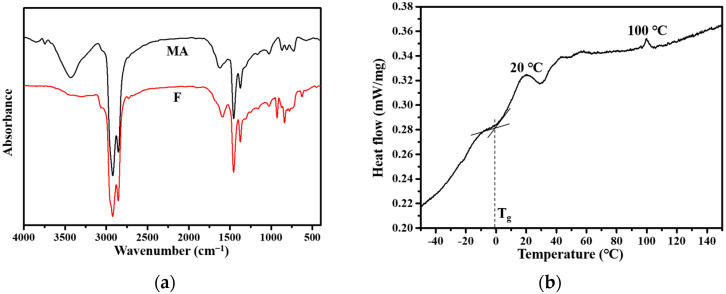
The FTIR spectrum (**a**) and DSC curve (**b**) of F.

**Table 1 materials-15-04346-t001:** Synthetic conditions of PPW and the corresponding PPWA.

PPW	Pyrolysis Conditions of PPW		PPWA
	Temperature (°C)	Pressure (MPa)	
**360-0.1**	360	0.1	**A**
**360-0.5**	360	0.5	**B**
**360-1.0**	360	1.0	**C**
**380-0.1**	380	0.1	**D**
**380-0.5**	380	0.5	**E**
**380-1.0**	380	1.0	**F**
**400-0.1**	400	0.1	**G**
**400-0.5**	400	0.5	**H**
**400-1.0**	400	1.0	**I**

**Table 2 materials-15-04346-t002:** SFE components of the probe liquids at 20 °C.

Probe Liquids	SFE Components (mJ/m^2^)				
	*γ^LW^*	*γ^+^*	*γ^−^*	*γ^AB^* ^1^	*γ* ^2^
Water	21.80	25.50	25.50	51.00	72.80
Formamide	39.00	1.92	39.60	19.00	58.00
Ethylene glycol	29.00	1.92	47.00	19.00	48.00

^1^ *γ^AB^* is polar acid base component; ^2^ *γ* is total SFE.

**Table 3 materials-15-04346-t003:** SFE components of limestone (mJ/m^2^).

Aggregate	SFE Components (mJ/m^2^)				
	*γ^LW^*	*γ* ^+^	*γ* ^−^	*γ^AB^*	*γ*
Limestone	143.22	0.0023	393.68	1.89	145.11

**Table 4 materials-15-04346-t004:** Contact angles (θ) and coefficient of variance (CV%) of MA and PPWA with three probe liquids.

Asphalt	Water		Formamide		Ethylene Glycol	
	θ (°)	CV (%)	θ (°)	CV (%)	θ (°)	CV (%)
MA	106.68	2.10	95.06	1.70	93.1	2.66
**F**	133.10	1.97	89.14	1.93	89.56	1.60

**Table 5 materials-15-04346-t005:** SFE parameters of MA and **F**.

Asphalt	SFE Components (mJ/m^2^)				
	*γ^LW^*	*γ* ^+^	*γ* ^−^	*γ^AB^*	*γ*
MA	20.08	0.42	2.71	2.15	22.23
**F**	33.44	0.25	6.59	2.55	35.98

## Data Availability

Not applicable.

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
