# Peer review of "Preparation of Wax-Based Warm Mixture Additives from Waste Polypropylene (PP) Plastic and Their Effects on the Properties of Modified Asphalt"

_materials, 2022, doi:10.3390/ma15124346_

Round 1
Reviewer 1 Report
The paper "Preparation of wax-based warm mixture additives from waste polypropylene (PP) plastic at different temperatures and pressures and their effects on the properties of modified asphalt" is within the scope of Materials. It could be interesting, but its form is not proper to be considered for publication.
Firstly, the English language must be revised (in the first line of the Abstract there is the first grammar error, and along the manuscript there are other errors).
The comparison between the modified asphalt and the natural one (as it seems in the abstract The product prepared at 380 °C and 1.0 MPa (380-1.0) can lead to a 61% penetration improvement and 46.8% viscosity reduction (135 °C) compared with matrix asphalt.) is a scientific error. It is not correct and it is misleading such comparison. We should compare N modified asphalt mixes (with different modifiers) to say if the tested one gives interesting results. Olympic and paralympic runners do not play the same game.
line 62: " exhibited better warm mixing performances." better than what?
Some information are not necessary in a scientific paper and should be omitted (e.g., plant in Chongqing, China. ... purchased from Shell Co., Ltd. (Foshan, China)... purchased from Sinopharm Chemical Reagent Co., Ltd. (China)).
Reviewer 2 Report
In the proposed study, the authors examined the influence of wax-based additives prepared by pyrolysis of waste polypropylene on the physical and rheological properties of the modified asphalt. Although the manuscript provides solutions for the reuse of waste plastic, some lacks are notable and should be corrected before its consideration for publishing. My specific comments are given below:
- In the introduction part, the authors should emphasize and state the novelty of this work.
- English must be improved. Besides, check and correct all typos throughout the manuscript.
- In line 76 provides additional information on did the reaction time was 20 min at the desired temperature or heating the rector and the reaction time was 20 min.
- Please uniform °C at Figures and text.
- Authors should compare their results and findings with the literature.
- Check the number of pages for reference 7.
- Indicate direction for future research and propose some improvements.
Reviewer 3 Report
I am have read this paper and found interesting. However this manuscript need to revise and more discussion is needed .
Please check my comments
- Introduction
- Between the line no. 39-40, it is difficult to find out the correlation between the WMa and PP.
- In line no. 48, it would be better to provide more explanation about the merit of high-hydrocarbon content in waste PP.
- Between the line no. 52-53, the need for research on the production of wax from waste plastic is not convincing. Furthermore, the subject suddenly has been changed without further explanation from PP to wax and again PP (in line 57).
- Results and Discussion
3.1. Softening point
- Between the line no. 118-121, the I sample was not followed the explained trend. Also, the effect of high pressure on pyrolysis needs to be explained in more detail (line 124-125).
3.2. Penetration
- Line from 131 to 132, needs an explanation for how the PPWs can reduce the viscosity and how the warm-mix additive can reduce the penetration of asphalt.
- Between the line no. 138-139, it is necessary to explain why the PPW prepared at excessive temp. and pressure does not affect the reduce in viscosity.
3.3. Ductility
- In line 157 and 158, the I sample didn’t followed the explained trend.
- The explanation between the line 158-161 seems not correct, because the influence of pyrolysis temp. and pressure on the properties of samples was not found only from the F sample, but also it can be observed from the “trend” of all samples except sample I.
3.4. Rotational viscosity
- In line from 165 to 171, it explained only the facts which showed in the Figures without demonstration of the mechanisms. In addition, after line 170, there is no further explanation about the reason why the wax-based WMa have melted at lower temp.
3.5. Rheological properties
- From line no. 183 to 187, there is a lack of explanation as to how the melting of PPW affects the rheological behavior, and also why F sample showed the highest G* value.
3.6. Adhesion properties
- It should be explained why only samples MA and F were selected for the measurement of adhesive properties before explaining the observed facts.
Reviewer 4 Report
- The title needs to be revised. It looks like a short abstract than the title, requires a short and simplified form.
- It seems like the Introduction section is too short and not completely describing the problems associated with the plastic materials in the industrial sector.
- The authors needs to include some sections in the Introduction section about the literature studies where what are the common plastics and additives commonly used in the industries for the product development, their affect onto the environment, their degradation capacity/behavior etc.
- In the Results and discussion section, the authors needs to provide at least preliminary characterization of their synthesized plastics like the morphological analysis, porosity, crystallinity, elemental composition analysis etc.
- Although the authors provided the analysis like softening point, penetration, ductility, rotational viscosity, rheological properties, and adhesion properties, I still feel like some strong analysis to publish in this journal is missing. If possible, the authors need to improve the manuscript by including some more characterizations and testing.
- Mechanical testing of the sample can be included wherever possible like Tensile strength, flexural strength, impact test, thermal stability, aqueous stability etc.
Reviewer 5 Report
In this paper, the authors contribute to the issue of the pyrolysis of waste plastics for wax production to improve the properties of modified asphalt. They prepared PPW from WPP via pyrolysis in a pressure reactor and then they obtained PPWA by blending PPW. Next, some tests have been processed (softening point, penetration, ductility, and rheological test) to highlight the properties of the PPWA. Conclusions are clear and consistent. The manuscript is interesting and well documented. The subject falls within the journal topic.
However, there some questions should be addressed:
- It is not clear from what the PPW acronym comes.
- Introduction, 41-42 lines: ‘As one of the representatives of plastics, polypropylene (PP) plastic is extensively used owing to its abundant source, low price, and chemical corrosion.’ What means ‘used owing to its … chemical corrosion’?
- Introduction, 46-47 lines: ‘However, landfill and incineration methods have the potential to pollute the surrounding soil and water’. Incineration pollutes the air.
- Introduction, 52-53 lines: ‘Few studies have been devoted to the production of waxes from waste plastic (PPW), particularly those used to modify asphalt [21-26].’ Authors should clear show what is new in their research comparing to the references cited.
Round 2
Reviewer 1 Report
The comparison between the modified asphalt and the natural one is a scientific error. It is not correct and it is misleading such comparison. We should compare N modified asphalt mixes (with different modifiers) to say if the tested one gives interesting results.
Author Response
Thanks for your question. We get your point now. The neat matrix asphalt seems like a reference in lots of studies. Please see articles “Constr. Build. Mate. 2018, 189, 882–889” “Constr. Build. Mate. 2016, 115, 294–298” “Materials 2019, 12, 1280-1296” and so on. The matrix asphalt with penetration grade of 70 is widely used in hot mix asphalt because of its excellent properties. So, it is necessary to compare the changes in performance before and after the additives, such as PPWs, are incorporated into the matrix asphalt.
Reviewer 3 Report
I am satisfied with the revised version and happy to accept in the current form
Author Response
Thanks a lot.
Reviewer 4 Report
I see that the authors made significant changes to my earlier commented paper. All the reviewed comments were addressed satisfactorily. Also they included some physical characterization of the prepared samples. Therefore, I must accept this paper in its present form.
Author Response
Thank you very much.